# Nature of power electronics and integration of power conversion with communication for talkative power

Xiangning He [1,2], Ruichi Wang [1,2], Jiande Wu [1✉] & Wuhua Li [1]

Power electronics and communication electronics are both based on electromagnetic theory, but they are usually regarded as two distinct subfields in electrical engineering. In fact, however, electric power is the most common matter-based carrier of messages; thus, power electronics and communications can be jointly considered. Here we study the essential nature of dc-dc power converters and characterize the similarity of their operation principle to that of communication systems. Based on this similarity and the double modulation methods used in power electronics and communication, a double modulation strategy for both power and data is presented and applied in dc-dc power converters to achieve what we call 'talkative power'. A modulation strategy called frequency hopping-differential phase shift keying (FH-DPSK) is also presented to overcome the crosstalk between chosen transmission systems. The proposed talkative power strategy sheds new light on and provides inspiration for the further development of power electronics and communication.

[1] College of Electrical Engineering, Zhejiang University, 310027 Hangzhou, Zhejiang Province, China. [2] These authors contributed equally: Xiangning He, Ruichi Wang. ✉email: eewjd@zju.edu.cn

Power electronics and communication electronics are both based on electromagnetic theory, but they are usually addressed separately as two distinct subfields of electrical engineering. Communications are usually considered merely in the sense of the abstraction of messages from the communication medium, while the matter-based carrier of these messages, which is usually electricity[1], is neglected. In communication theory, electricity itself is no longer the phenomenon of interest; rather, it is useful only as a means of transmitting information[2]. As Norbert Wiener said, "Information is information, not matter or energy"[1,3]. This opinion makes sense from some perspectives but completely ignores the connections between communications and electric power. On the other hand, communication is indispensable in a practical electrical system to achieve distributed power control for modularization, intellectualization and plug-and-play functionality; however, the communication mechanism is usually designed separately, with an independent structure[4–6]. Nevertheless, to enhance compactness and decrease costs, power line communication technology has been proposed and widely adopted[7,8]. The combination of power transfer and communications has also been investigated in the wireless field, where it is known as simultaneous wireless information and power transfer[1]. In these techniques, electrical signals are considered from the perspectives of both power and communication. However, most efforts have focussed on the methods of power and data transmission and the receiver architecture, with separate approaches for generation and coupling[9–11].

In communication systems, there are three basic elements: transmitters, channels, and receivers. A transmitter and a receiver are located at two separate points in space, and the physical medium that connects them is called a channel. A transmitter mainly consists of a data modulator, and a receiver mainly consists of a data demodulator. Therefore, the communication process can be essentially divided into data modulation, transmission and demodulation. The base-band signal is modified to a high-frequency carrier to make it suitable for transmission. The modulated signal is then transmitted over the channel to the receiver. Finally, the receiver recreates the original signal, through a process known as demodulation.

In this paper, we reconsider power electronic converters from an interdisciplinary perspective and propose a method of integrating communication into direct current (dc)–dc converters to achieve what we call 'talkative power'. With the proposed integration method, it becomes possible for power to 'talk', or communicate, during conversion. The 'talkative power' converter has many applications in contexts such as distributed power electronic systems, modularized dc devices, luminaires and the Internet of Things, which are dual-purposed to provide both energy and communications currently in separate ways. In addition, a novel modulation method called frequency hopping–differential phase shift keying (FH-DPSK) is proposed along with the talkative power strategy to achieve effective communication while avoiding interference between power conversion and data transmission, which will be valuable for supporting new, advanced techniques in conventional light-emitting diode (LED)/lighting technology. A fundamental and rigorous analysis is presented based on this innovative and general concept, revealing the natural relation between the two subfields of power electronics and communications and providing inspiration for their further joint development.

## Result

### Essential nature of power converters for communication.
In a power electronic converter, a reference signal is modulated to a switching frequency to serve as the gate signal for one or more switches by a modulator and is then amplified by an input power source and the switch(es) (in some cases, passive elements are also involved). The modulated and amplified signal is then demodulated to obtain a signal with the required level of output power and the same form as the reference (usually either a dc form or a power frequency alternating current (ac) form). Accordingly, the power conversion process can be divided into distinct stages of modulation, power amplification and demodulation, which are, in some sense, analogous to the stages of the communication process. Transmission is ignored in this case since the modulator and demodulator are located at the same point in space.

There are four kinds of power electronic converters, namely, dc–dc power converters, dc–ac inverters, ac–dc rectifiers and ac–ac power converters, among which dc–dc converters are the most basic. Various dc–dc topologies have been designed to meet different application requirements, such as buck converters for voltage step-down, boost converters for voltage step-up and buck–boost converters for inverse voltage step-up and step-down. Most common dc–dc converters can be divided into two categories based on their operation principles: a periodic square wave sequence is generated before the power demodulator, which is either (i) an LC low-pass filter (LPF), as in the cases of buck, Ćuk and Zeta converters, or (ii) an envelope detector (a series-connected diode followed by a shunt-connected capacitor), as in the cases of boost, buck–boost and single-ended primary-inductor converters. On the load side, a dc output voltage is achieved, the amplitude of which is determined by the input voltage and the duty cycle of the gate signal[12]. Because of the limited capacity of the inductors and capacitors in practice, the switching ripples cannot be completely filtered out. However, in a well-designed dc–dc converter, the amplitude of the residual switching ripple should be <1% of the output dc component[12].

In power electronics, modulation is adopted to achieve controllable power conversion. Considering the operation principle of dc–dc converters, in which the modulated signal is used as the gate signal for the switch(es) and both the level and form of the output voltage should be controllable, the modulation process is subject to two constraints: (1) the switches should operate in either an on or off state to reduce power losses, meaning that only pulse modulation is applicable, and (2) the power output level is determined by the time durations of the on and off states of the switch(es), meaning that the upper and lower limits on the durations of the pulses should be adjustable. Based on these considerations, pulse width modulation (PWM) is the best choice. In 1964, Schonung et al. introduced PWM into power electronics for dc–ac inverter control[13], and since then, PWM has become the most important and mainstream modulation method in power converters. In this method, a reference signal is compared with a triangle or sawtooth wave sequence to generate a PWM sequence as the gate signal for the switch(es). The input voltage and the duty cycle of the PWM sequence determine the output voltage level, while the envelope of the PWM sequence determines the output voltage form.

To further investigate PWM in a dc–dc converter, the corresponding analysis starts with the sampling process. The sampling process (Fig. 1a–c) is impulse amplitude modulation process[14], in which a periodic sequence of impulses is used to multiply the original signal in the time domain[15]. In practice, it is almost impossible to generate an ideal impulse sequence; therefore, this sequence is usually replaced with a periodic pulse sequence generated by a sample-hold circuit[15]. Therefore, the practical sampling process (Fig. 1d, e) is pulse amplitude modulation (PAM) process. PWM (Fig. 1f) is another pulse

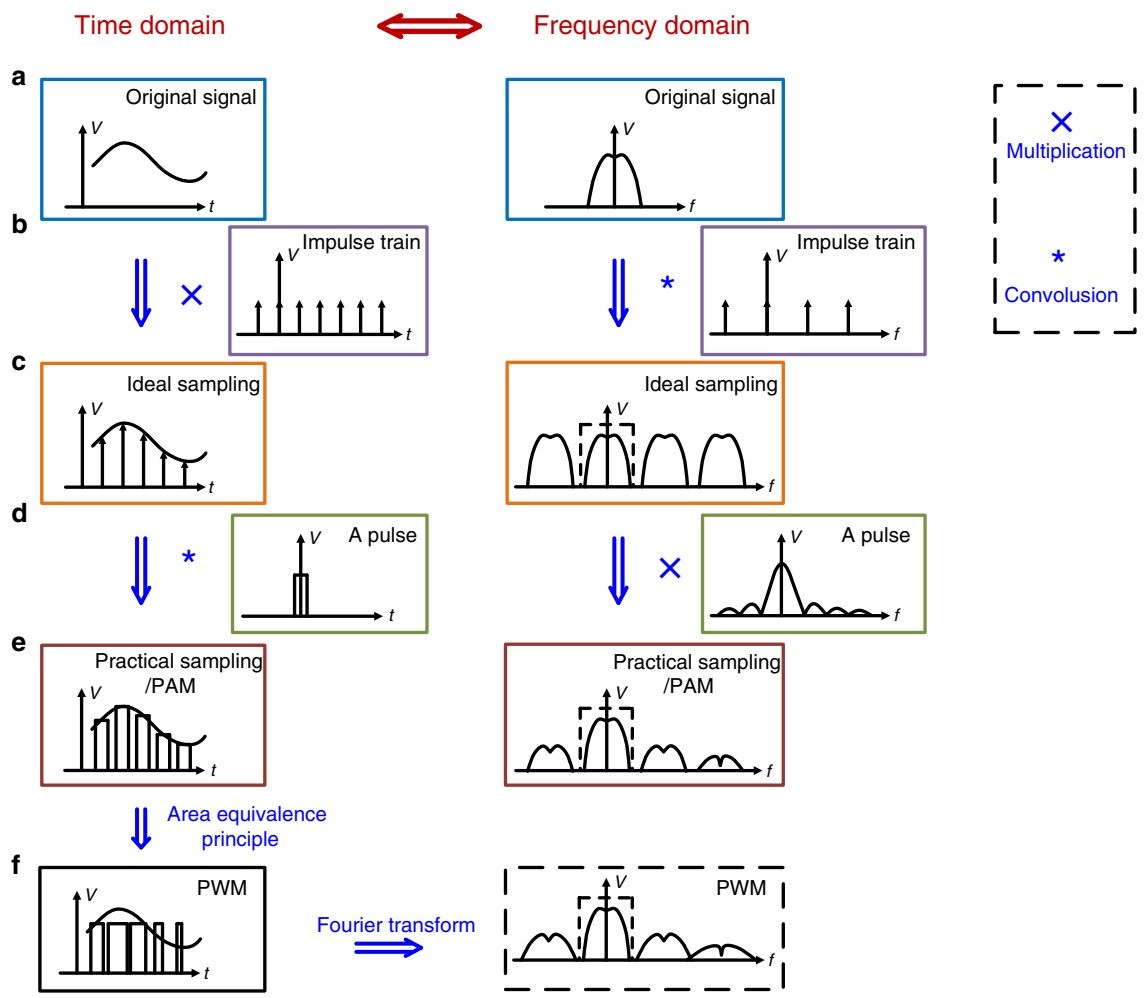

**Fig. 1 Waveforms and spectrum corresponding to sampling and modulation.** This figure shows the relationship between sampling, PAM and PWM. Both time-domain waveforms and frequency-domain spectrum are depicted. **a** The continuous original signal to be sampled or modulated. **b** An impulse train, which is a sequence of Dirac functions and can be employed as a sampling function. **c** The ideal sampling result, which is the result of the multiplication of **a** and **b** in the time domain and convolution in the frequency domain. **d** A pulse in the time domain and its corresponding spectrum. **e** The sampling result obtained in practice, in which case the periodic sequence of impulses (Dirac functions) is replaced with a sequence of pulses of equal width. This process is also known as PAM. This result is derived through the convolution of **c** and **d** in the time domain and multiplication in the frequency domain. **f** The result of PWM—derived from **e** in the time domain based on the area equivalence principle.

modulation method, expressed as:

$$s_{\text{PWM}}(t) = K_{\text{PWM}} \sum_{k=-\infty}^{\infty} [u(t - kT_s) - u(t - kT_s - x_k T_s)] \quad (1)$$

In this equation, $K_{\text{PWM}}$ is the amplitude of the PWM sequence, $u(t)$ is the unit step signal, $T_s$ is the period of the PWM sequence and $x_k$ $(0 \leq x_k \leq 1)$ is the duty cycle in the $k$th period. Following the approach in ref. [16], the spectrum of a uniform-sampling trailing-edge PWM signal can be derived as:

$$S_{\text{PWM}}(f) = \frac{1}{j2\pi f} \sum_{k=-\infty}^{\infty} \left( e^{-j2\pi fkT_s} - e^{-j2\pi f(kT_s + x_k T_s)} \right) = \frac{-1}{j2\pi f} \sum_{k=-\infty}^{\infty} e^{-j2\pi fkT_s} \sum_{n=1}^{\infty} \frac{(-j2\pi f)^n}{n!} (x_k T_s)^n$$

$$= \frac{-1}{j2\pi f} \sum_{n=1}^{\infty} \frac{(-j2\pi fT_s)^n}{n!} \sum_{k=-\infty}^{\infty} e^{-j2\pi fkT_s} (x_k)^n = \frac{-1}{j2\pi f} \sum_{n=1}^{\infty} \frac{(-j2\pi fT_s)^n}{n!} F_n \left( e^{-j2\pi fT_s} \right)$$

$$= \sum_{k=-\infty}^{\infty} \sum_{n=1}^{\infty} \frac{(-j2\pi fT_s)^{n-1}}{n!} S_n(f - kf_c)$$

$$(2)$$

where $F_n \left( e^{-j2\pi fT_s} \right)$ is the discrete-time Fourier transform (DFT) of $(x_k)^n$ and $S_n(f)$ is the Fourier transform of $(x(t))^n$. As seen by

comparing Fig. 1e, f, PWM has an effect similar to that of PAM, especially near the fundamental frequency of the spectrum. Therefore, PWM is approximately equivalent to the sampling process near the fundamental frequency.

During the sampling process, recreation of the original signal is usually achieved by means of an LPF. Envelope detection is a basic and simple demodulation method that can be used in power demodulation. Therefore, in a dc–dc converter, power demodulation can be regarded as a process of signal recreation after sampling, in which case an LPF is adopted, or as a process of signal demodulation, in which case an envelope detector is adopted, consistent with the previous structure analysis of dc–dc converters.

A buck/boost converter is taken as an example for further analysis. Such a converter allows bidirectional power flows and hence always operates in continuous conductive mode. For a dc–dc converter operating in *dis*continuous conductive mode, in which the amplitude of the switching ripple is related to the output power, the proposed method is also applicable, but the receiver must be adaptable to a wide range of signal strengths. The structure of the buck/boost converter consists of a dc

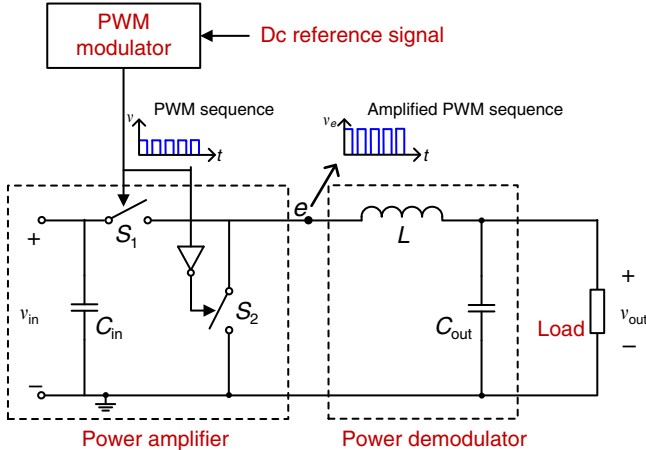

**Fig. 2 Structure of a buck/boost converter.** $S_1$ and $S_2$ are two identical metal-oxide-semiconductor field-effect transistors (MOSFETs) as ideal switches, $C_{in}$ denotes the input capacitor and $C_{out}$ denote the output capacitor. $L$ denotes the inductor. $v_{in}$ and $v_{out}$ are input and output voltages, respectively. The buck/boost converter can be divided into five components, as labelled in red: a dc reference signal, a PWM modulator, a power amplifier, a power demodulator, and a load. The power amplifier includes an input dc source and two switches operating in a complementary mode. The power demodulator includes an LC LPF. The waveform at point 'e' is the amplified PWM sequence.

reference signal, a PWM modulator, a power amplifier, a power demodulator and a load, as shown in Fig. 2. A PWM signal is generated by comparing the dc reference signal with a triangle or sawtooth wave sequence and is used as the gate signal for the switches. At point 'e' in Fig. 2, an amplified PWM sequence is achieved, with voltage levels of 0 and $v_{in}$. After the LC LPF, a dc power output signal is achieved, superimposed with a residual switching ripple.

**Integration of power electronics and communications.** According to the analysis presented above, both the communication and power conversion processes employ modulation and have similar structures and operation principles. In a buck/boost converter, a dc reference signal is modulated and the resulting PWM signal is amplified; this process corresponds to the transmission process in a communication system. The amplified PWM signal is then demodulated to power the load; this process resembles the demodulation process in a communication system. This similarity between the operation principles of communication and power conversion affords the possibility to integrate communications into a buck/boost converter. In modulation theory, a procedure in which a signal is modulated twice is a standard practice known as double modulation. In both communications and power electronics, double modulation is widely adopted to enhance system performance[17–20]. Based on the relation between power conversion and communications and the double modulation strategies applied in both areas, a double modulation method for power and data is proposed here. When this strategy is applied in a dc–dc converter, talkative power is achieved. In a buck/boost converter, data modulation is superimposed with power PWM. These two modulations should be independent and uncoupled so that the power conversion and communication processes will not influence each other. Specifically, with the superimposition of data modulation, the amplitude and duty cycle of the gate signal should remain unchanged. Therefore, only angular modulations, such as frequency

modulation and phase modulation, are applicable. Here we adopt orthogonal continuous-phase binary frequency shift keying (2FSK)[21] as the data modulation method in a wired power electronic system. In wireless power transfer (WPT) systems, the efficiency of power transfer strongly relies on a resonant operating frequency[22,23], while the spectrum of the information signal is no longer unimodal. Therefore, although this double modulation strategy can also be adapted to WPT systems, doing so will require further analysis and improvements.

The buck/boost converter includes a power demodulator in the form of an LC LPF. Then the dc and residual switching frequency components are transferred to the load for power transfer and to the receiver(s) for data demodulation. Thus power and data modulation are combined, where the power signal is also the carrier of the data; as a result, power and data signals can be transmitted over a common power line. The load and receiver(s) can be integrated or separate, but they must be connected to the buck/boost output power line. Each receiver consists of a signal conditioning circuit to extract the switching frequency component for data demodulation. Under these conditions, data are embedded into and transmitted with the power signal; thus the output power signal can 'talk' to any device connected to the output power line, thereby achieving talkative power.

Double modulation methods for power and data can also be developed for other dc–dc topologies by applying a similar operation principle. The dc reference is modulated to a switching frequency via PWM, and frequency- or phase-based data modulation is superimposed. The modulated signal is then amplified by the input dc source and the switch(es) (some passive elements may also be involved). Subsequently, the dc voltage is recreated/demodulated by either an LC LPF or an envelope detector and is then transmitted to the load and receiver(s). In this way, for any dc–dc converter in which the proposed double power-and-data modulation method is implemented, its output can be used not only to power a load but also to communicate with other devices.

The Fourier transform of a single period of a PWM sequence is:

$$|s(n\omega_0)| = \left| \frac{E}{\pi} \frac{\sin(\pi n d)}{n} \right| \quad (3)$$

where $E$ is the amplitude of the PWM wave, $d$ is the duty cycle, $\omega_0$ is the angular frequency (in radians) of the PWM sequence and $n$ is the harmonic order. The fundamental amplitude decreases as $|d - 0.5|$ increases; that is, an extreme (wide/narrow) duty cycle leads to a lower fundamental amplitude. In addition, an extreme duty cycle results in a low power-conversion efficiency[24]. Fortunately, in a well-designed dc–dc converter, an operating state with an extreme duty cycle can be avoided.

In practice, a buck/boost converter is usually a closed-loop output voltage control system composed of a proportional-integral (PI) compensator $G_c(s)$, a PWM modulator $G_m(s)$, a buck/boost converter $G_{vd}(s)$ and a voltage sampling feedback loop $H(s)$, all connected in series. Taking output voltage control as an example, the open-loop transfer function is:

$$G_0(s) = G_c(s) \cdot G_m(s) \cdot G_{vd}(s) \cdot H(s)$$
$$= \left( K_p + \frac{K_i}{s} \right) \cdot \frac{1}{V_M} \cdot \frac{V_{in}}{1 + s\frac{L}{R} + s^2 LC} \cdot \frac{R_b}{R_a + R_b} \quad (4)$$

where $K_p$ and $K_i$ are PI parameters; $V_M$ is the amplitude of the triangle wave used in PWM generation; $V_{in}$ is the input dc voltage; $L$ and $C$ are values of the inductor and the capacitor of the LC LPF, as shown in Fig. 2; $R$ is the load resistance and $R_a$ and $R_b$ are the output sampling resistances of the voltage divider $\left( V_{sample} = \frac{R_b}{R_a + R_b} V_{out} \right)$. The cut-off frequency of the control loop

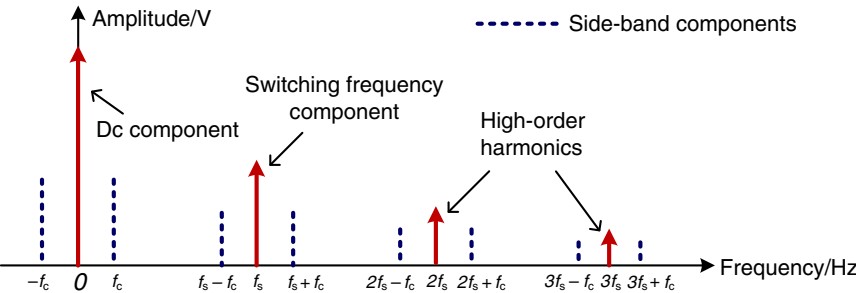

**Fig. 3 Output voltage spectrum of a buck/boost converter with PI-compensated output voltage control.** The red components are the dc component, the switching frequency component and the high-order harmonics. $f_s$ denotes the switching frequency and $f_c$ denotes the cut-off frequency of the control loop. The dotted blue lines are the side-band components induced by the PI-compensated voltage control loop. They are distributed on both sides of the red components at frequency intervals equal to the cut-off frequency of the control loop.

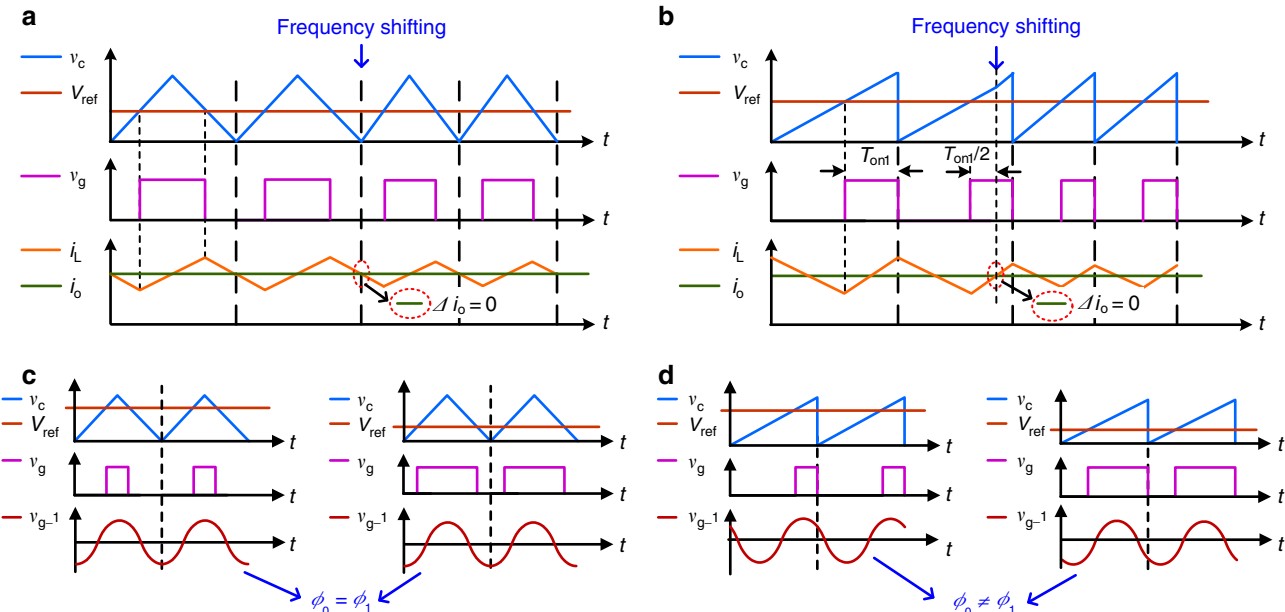

**Fig. 4 Comparison of two different carriers of a triangle wave and a sawtooth wave. a** A triangle wave sequence is adopted to generate the PWM sequence. The top waveforms, $v_C$ and $V_{ref}$, are the triangle wave and the dc reference signal, respectively. The middle waveform, $v_g$, is the generated PWM sequence, i.e., the gate signal. The bottom waveforms, $i_L$ and $i_o$, are the inductor and output currents, respectively. At the moment of the frequency shift at the peak of the triangle wave, the output current remains stable ($\Delta i_o = 0$). **b** A sawtooth wave sequence is adopted to generate the PWM sequence. The top waveforms, $v_C$ and $V_{ref}$, are the sawtooth wave and the dc reference signal, respectively. The middle waveform, $v_g$, is the generated PWM sequence, i.e. the gate signal. The bottom waveforms, $i_L$ and $i_o$, are the inductor and output currents, respectively. At the moment of the frequency shift, the output current remains stable ($\Delta i_o = 0$). The frequency shift occurs on the upward ramp of the sawtooth wave, where the high level of $v_g$ lasts for ½$T_{on1}$. **c** A triangle wave sequence is adopted to generate the PWM sequence. The top waveforms, $v_C$ and $V_{ref}$, are the triangle wave and the dc reference signal, respectively. The middle waveform, $v_g$, is the generated PWM sequence, i.e. the gate signal. The bottom waveform, $v_{g\_1}$, is the fundamental component of the gate signal. When the duty cycle of the gate signal changes, the phase of $v_{g\_1}$ remains constant. **d** A sawtooth wave sequence is adopted to generate the PWM sequence. The top waveforms, $v_C$ and $V_{ref}$, are the sawtooth wave and the dc reference signal, respectively. The middle waveform, $v_g$, is the generated PWM sequence, i.e. the gate signal. The bottom waveform, $v_{g\_1}$, is the fundamental component of the gate signal. $\Phi_0$ and $\Phi_1$ refer to the phase of before and after the duty cycle changes. The phase of $v_{g\_1}$ varies with a change in the duty cycle.

is $f_c$. The spectrum of the output voltage is shown in Fig. 3. The dotted lines represent the side-band frequencies induced by closed-loop output voltage regulation, which are equal to $nf_s \pm f_c$, where $n$ is an integer. In PWM/2FSK modulation, two frequencies, $f_0$ and $f_1$, are adopted as the switching frequencies to achieve 2FSK modulation. Each frequency occupies a bandwidth of $2f_c$ with its centre at $f_0$ or $f_1$, respectively. No overlap of the switching frequency and the frequency component introduced by the control loop is allowed since this may lead to

misrecognition of the switching frequency during data demodulation. Therefore, the interval between the two switching frequencies should satisfy the constraint $|f_1 - f_0| > 2f_c$. For other cases of closed-loop control, such as output current control, this constraint remains the same. Fortunately, for most common dc–dc converters, the switching frequencies $f_1$ and $f_0$ are much higher than the cut-off frequency of the control loop. Therefore, the power control loop has little influence on the selection of the switching frequencies in the case of double power-and-data

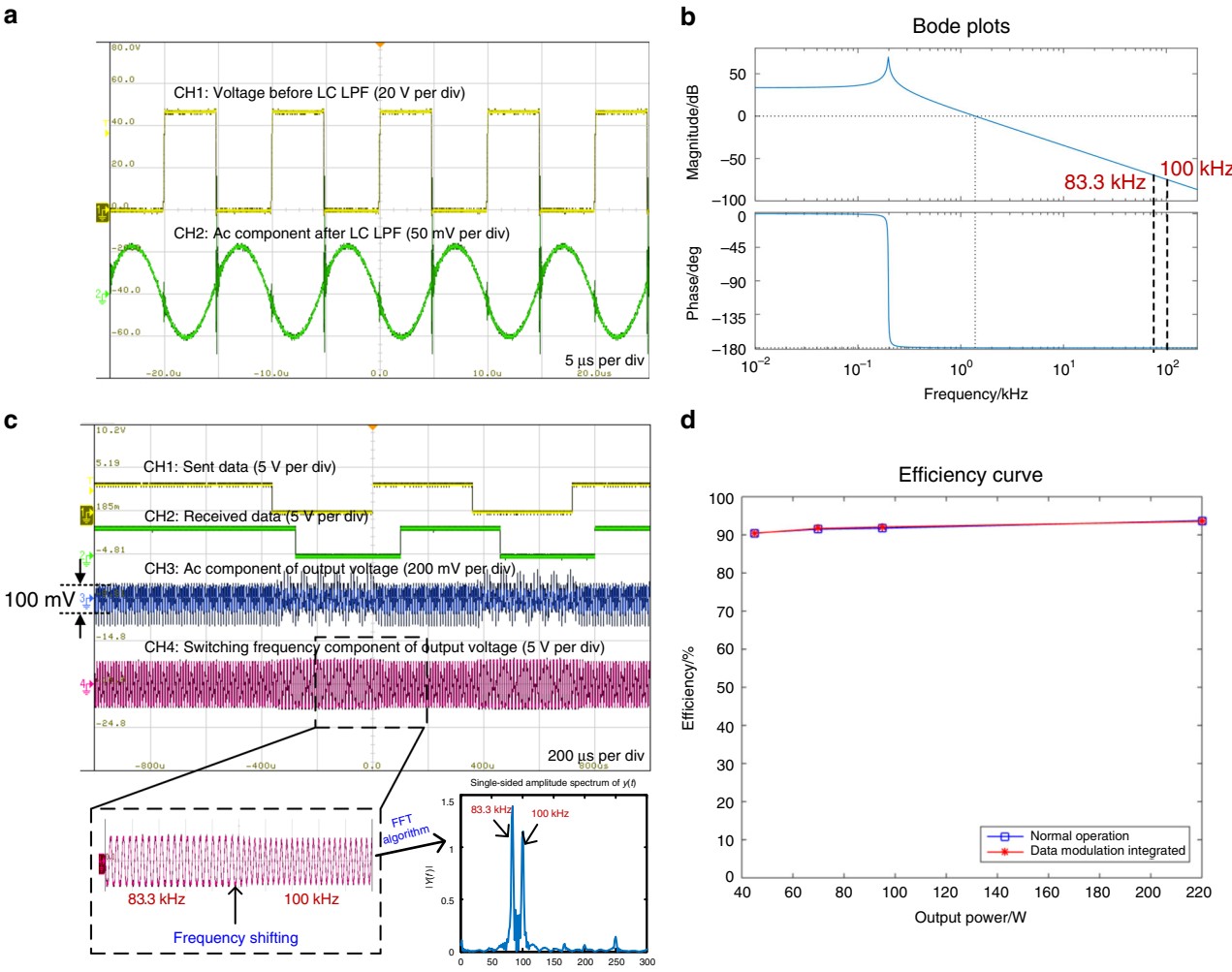

**Fig. 5 Experimental results of double PWM/2FSK modulation for power and data based on a conventional buck/boost converter. a** The buck/boost converter operating at 100 kHz. CH1 is the voltage waveform before the LC LPF. It is a square wave sequence with a fundamental frequency of 100 kHz. CH2 is the ac component of the output voltage. It is approximately a sinusoidal waveform with a fundamental frequency of 100 kHz. After the LC LPF, the switching frequency component is attenuated, and the phase shift is approximately 180°. **b** Bode plots for the LC LPF with a load $R$ at switching frequencies of 100 and 83.3 kHz; the amplitude is attenuated, and the phase shift is approximately 180°. **c** Four oscilloscope waveforms representing the sent data, the received data, the ac component of the output voltage and the switching frequency component of the output voltage. A view of the switching frequency component of the output voltage waveform that has been magnified on the time axis is shown, and its fast Fourier transform (FFT) is a single-sided spectrum, with peaks at 83.3 and 100 kHz. **d** Efficiency curves based on different load conditions. The blue curve represents normal operation conditions, and the red curve represents the case with integrated data modulation. The source data are provided as a source data file.

modulation. In addition, the variation of the switching frequencies for communication has little influence on power control. It is known that the voltage regulation performance of a power converter is mainly determined by the control loop bandwidth rather than the switching frequency (given that the switching frequency is at least five times of the bandwidth). A converter (buck, boost, etc.) with the proposed communication function is identical to one without this function in terms of voltage regulation performance as long as their control loop bandwidths are the same and well designed. For communication purposes, to minimize the inter-symbol interference, the switching frequencies $f_1$ and $f_0$ should be orthogonal throughout the duration of a symbol, i.e. $f_i = (n_c + i)k\nu_b$, where $i = 0$ or 1, $k$ and $n_c$ are constant integers, and $\nu_b$ is the baud rate. In addition, the switching frequencies should be selected to satisfy the requirements from the power perspective, such as switching losses and output power quality.

Note that at the moment of the switching frequency shift, the output current may exhibit a small distortion, which is

undesirable for power conversion[25,26]. To avoid this distortion, the switching frequency shift should be selected to occur at the average-current-crossing instant of the inductor. The PWM sequence is generated by comparing the dc reference signal with a triangle or sawtooth wave sequence. For the case of a triangle wave sequence, the moment of the frequency shift should be selected to lie at the peak of the triangle wave, as shown in Fig. 4a. However, for a sawtooth wave sequence, the moment of the frequency shift should be selected to lie on the upward ramp, and the exact time should be calculated according to the duty cycle, as shown in Fig. 4b. Consequently, for PWM/2FSK, a triangle wave sequence is the preferred choice for ease of implementation.

With the adoption of PWM/2FSK modulation, the two switching frequencies $f_0$ and $f_1$ are used to represent data values of 1 and 0. The converter performance will not be degraded if a conservative design is employed, meaning that the filter design is based on the lower frequency, $f_0$, while the thermal design is based on the higher frequency, $f_1$. Double modulation consisting

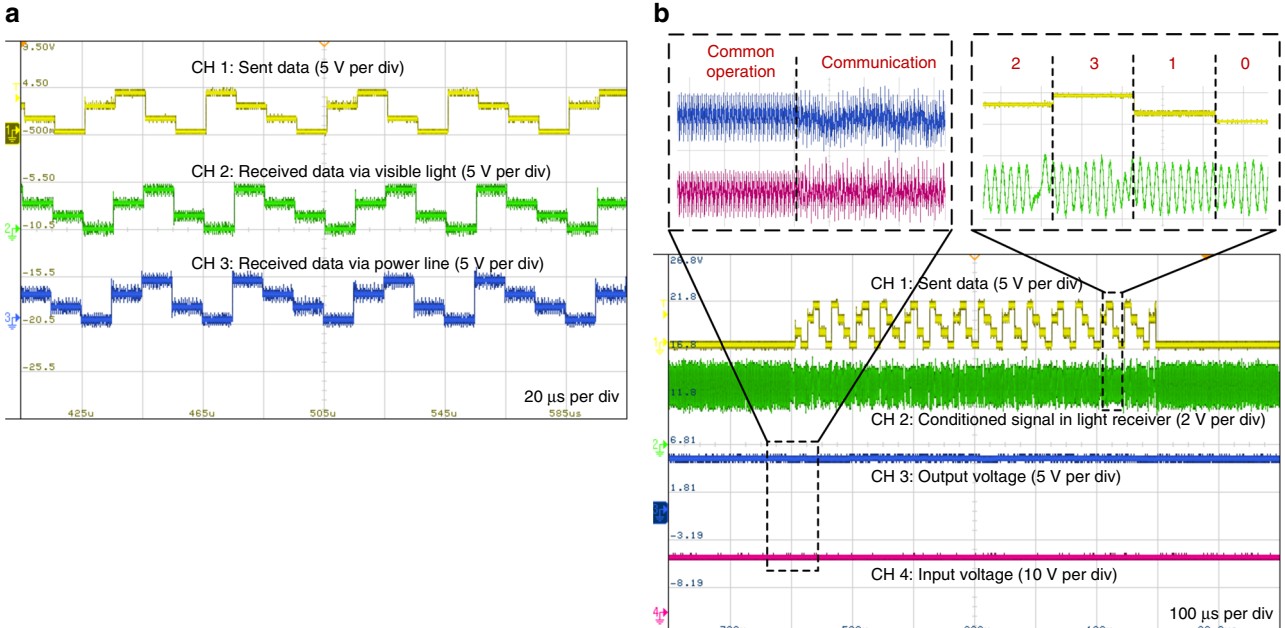

**Fig. 6 Experimental results based on a GaN MOSFET buck/boost converter in LED lighting system.** This experiment is based on PWM and FH-4DPSK modulation for PLC and VLC. **a** The communication waveforms of the sent data (CH1), the data received via visible light (CH2) and the data received via the power line (CH3). **b** The waveforms of sent data (CH1), conditioned signal in light receiver (CH2), output voltage (CH3) and input voltage (CH4). Zoomed views of the waveforms are the conditioned signal and the input/output voltages in the common operation and communication states.

of a combination of PWM and 2PSK can also be adopted. In this case, the two phases 0 and $\pi$ represent digital data values of 0 and 1, respectively. As shown in Fig. 4c, d, to ensure decoupled power and communication control, only a triangle wave should be adopted in PWM/2PSK modulation[27]. PWM/2PSK modulation can be implemented for a standalone converter. However, in a distributed power system with parallel converters, components at the same frequency from other converters will be a source of serious interference in communication, causing the signal-to-noise ratio (SNR) to deteriorate. To solve this problem, a novel FH-DPSK modulation scheme is proposed here. In this scheme, the converter operates at a frequency of $f_0$ in the normal state but at a frequency of $f_1$ in the transmitting state, and the data values are represented by the phase of $f_1$. The power conversion and data transmission frequencies are assigned to different bandwidths to avoid interference. To further increase the communication rate, M-ary modulation can be employed, in which a phase shift may lead to variation of the switching frequency. Similar to the case of PWM/2FSK modulation, a conservative design is again adopted to ensure the performance of the converter. The detailed analysis of PWM/FH-DPSK modulation is similar to that of PWM/2FSK modulation and therefore is not presented in this paper.

The switching ripples of other dc–dc converters act as noise in the communication channel. Similarly, the white noise present in a communication channel also consists of the superposition of the white noise from each dc–dc converter. Owing to the orthogonal nature of the switching frequencies, other switching ripples will have little influence on the demodulation performance, but the white noise will lead to a lower SNR. Therefore, the maximum number of dc–dc converters that can be connected to a common power line is determined by the required communication quality.

For the design of power electronic converters, harmonic problems should also be considered. Data modulation may enrich the frequency components of the switching harmonics, but the total harmonic power will not increase. If spread spectrum data modulation is adopted, the switching harmonics will be spread and suppressed in the frequency domain. Thus the harmonics can be mitigated without affecting power conversion performance.

**Experimental verification**. To verify the proposed talkative power strategy, an experiment using double PWM/2FSK modulation was first performed based on a conventional buck/boost converter. In the structure depicted in Fig. 2, the input voltage was set to 48 V, and the output voltage was 24 V; thus the duty cycle was $d = 0.5$. The two switches were MOSFETs. For the LPF, $L = 650\ \mu H$ and $C = 1\ mF$, thus the cut-off frequency was 127 Hz. PI-compensated closed-loop output voltage control was adopted, with discrete PI parameters of $K_p = 2$, $K_i = 1 \times 10^4$ and $T_s = 20\ \mu s$. The cut-off frequency of the power control loop was $f_c = 2.3\ kHz$. Two switching frequencies of 100 and 83.3 kHz (with periods of 10 and 12 μs, respectively), with adjacent intervals larger than $f_c$, were adopted. $f_1 = 100\ kHz$ represented a data value of 1 and the idle state, while $f_0 = 83.\dot{3}\ kHz$ represented a data value of 0. The baud rate was set to 2.78 kB (240 μs for one symbol), and the bit rate was 2.78 kbps. The two frequencies were orthogonal in one bit, and their phases were continuous.

Figure 5a shows the voltages before and after the LC LPF. The switching frequency component is attenuated, whereas on the output side, the amplitude of the residual switching frequency component is approximately 100 mV (0.42% of the output dc voltage), which satisfies the output power quality requirement for a dc–dc converter in most applications[12] while also being significant for communications. Figure 5b shows the Bode plots of the LC LPF with a load $R$, from which it can be seen that the phase shift is 180° at the switching frequencies. Because of the equivalent series resistance of the capacitor, the amplitude attenuation is reduced compared with that in the theoretical analysis, and the phase shift deviates from 180°. Therefore, for phase-continuous frequency modulation, the output phases of the

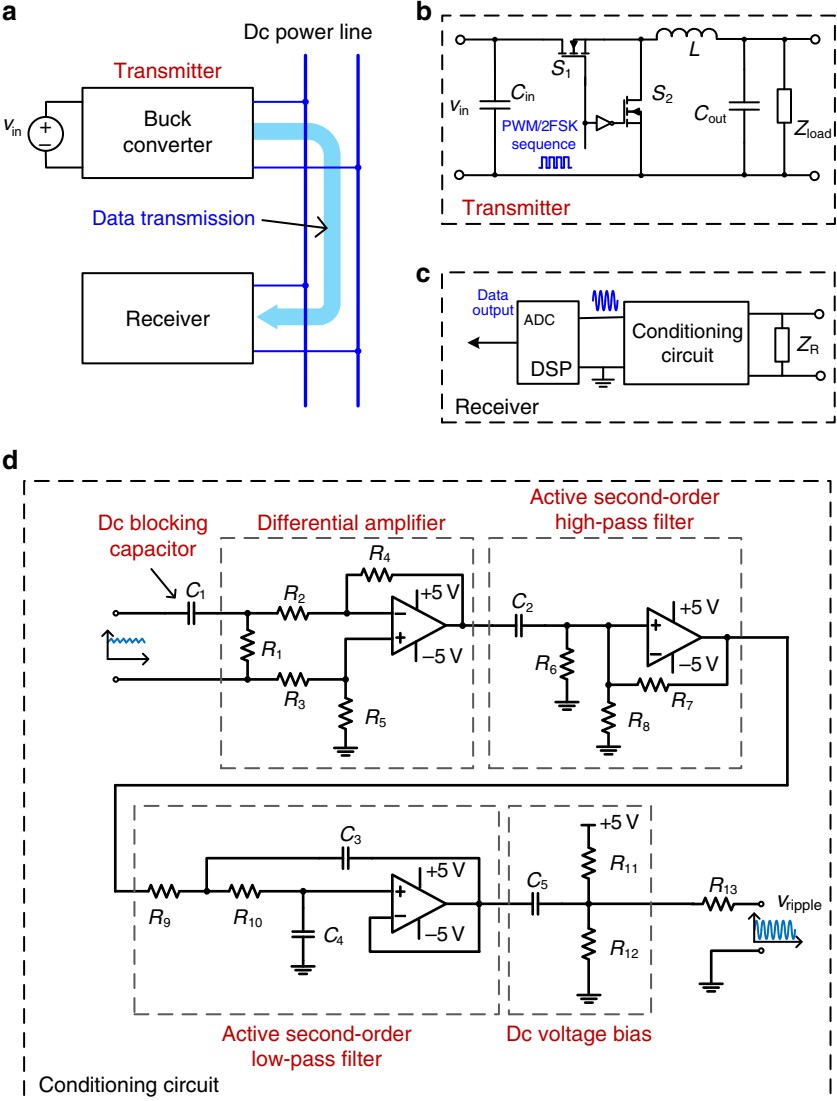

**Fig. 7 The structure of the prototype used in the basic validation experiment. a** The general structure of the prototype. It includes a buck/boost converter as a data transmitter and a receiver sharing a common power line with the buck/boost converter. $v_{in}$ denotes the input voltage of the buck converter. Data are transmitted from the buck/boost converter to the receiver via the dc power line. **b** Detailed circuit of the buck/boost converter, in which $v_{in}$ denotes the input voltage, $C_{in}$ denotes the input capacitor while $C_{out}$ denote the output capacitor. $L$ denotes the inductor and $Z_{load}$ denotes the load. The modulated signal is the gate signal for MOSFET $S_1$, and its complementary signal is the gate signal for MOSFET $S_2$. **c** The structure of the receiver. It consists of load $Z_R$, a signal conditioning circuit and a DSP with analogue-to-digital converter (ADC) to output digital data. **d** The signal conditioning circuit in the receiver. This circuit is composed of a dc blocking capacitor, a differential amplifier, an active second-order high-pass filter, an active second-order LPF and a dc voltage bias circuit. $R_1$–$R_{13}$ denote resistors, and $C_1$–$C_5$ denote capacitors. $v_{ripple}$ denotes the conditioned signal voltage. The input signal, which has various frequency components and a large dc bias, is conditioned, and at the output port, a signal with a single frequency (a sinusoidal wave) and a proper dc bias is achieved.

switching frequency components are also continuous, while for differential phase modulation, the absolute phase is not important; thus both types of modulation are applicable. Figure 5c shows the digital data sequence that is sent and the amplified PWM waveforms after the LC LPF. The two frequencies 100 and 83.3 kHz represent digital data values of 1 and 0, respectively. On the load side, the dc component is dominant, while most of the switching ripple is filtered out. In the buck/boost converter, power modulation and demodulation are performed, and the required dc output voltage is available to power the load. The data are modulated and transmitted to the receiver via the output power line. CH4 in Fig. 5c is the waveform after conditioning at the receiver. The fundamental components of the switching

frequencies are conserved, while high-order harmonics are attenuated. After DFT demodulation, the original data can be recovered (CH2 in Fig. 5c). Figure 5d shows the efficiency curves measured under both normal operation conditions and communication-integrated conditions. The efficiency is almost the same under both conditions, further verifying that, with our proposed integration method, no additional power is consumed for communication.

We also applied the double modulation strategy in an LED lighting system, using a converter with GaN MOSFETs as the LED driver. The proposed double PWM/FH-4DPSK modulation scheme was adopted in the LED lighting system for integrated communication via both power lines and visible light. This

strategy based on a talkative power converter theoretically differs from the conventional visible light communication methods presented in the literature for luminaires[28–30], and there is no LED flickering, nor any interference between power conversion and data transmission. The experimental results are shown in Fig. 6a, b, where the switching frequency is 1 MHz. The four levels represent quaternary data signals of 0, 1, 2, and 3. The baud rate is equal to 100 kB, and the bit rate is 200 kbps for the talkative power converter. Further experimental details, including a description of the prototype system and its parameters, are provided in the "Methods" section.

In addition to LED lighting systems, the proposed double modulation strategy can also be applied in many other distributed power systems, such as battery management systems and optimizer-structured photovoltaic systems.

## Discussion

We have studied the essential nature of power electronic converters based on communication theory and established an analogy between the operation principles of power electronics and communications. On this basis, we have proposed a method of integrating communication into dc–dc converters to achieve what we call 'talkative power' in power electronic systems. This fundamental research offers a new perspective on power electronics. Combining power electronics and communications reveals the inherent links between these two subfields. The proposed talkative power strategy simplifies the communication structure in a dc–dc converter system and offers an incentive for further investigation of the relation between power electronics and communications. The proposed strategy has been adopted and experimentally verified in a wired dc–dc converter system. As the switching speeds of devices increase, more advanced data modulation methods can be employed, thereby greatly improving the communication rate.

## Methods

**Basic validation experiment**. In the first experiment, the PWM/2FSK method was adopted to achieve double power-and-data modulation based on a conventional buck/boost converter. A receiver was connected to the output power line of the buck/boost converter, as shown in Fig. 7a–d. In this experiment, the output voltage was controlled, and a PI voltage compensator was used. The parameters of the conventional buck/boost converter are listed in Table 1. Now, let us analyse the principles governing switching frequency selection. The two switching frequencies used for data modulation should be phase-continuous and orthogonal; thus these two frequencies and the bit rate should satisfy Eq. (5), where $v_b$ is the baud rate:

$$f_i = (n_c + i)kv_b, \quad i = 0 \text{ or } 1, \, n_c \text{ and } k \text{ are constant integers and } n_c \text{ and } k \geq 1. \quad (5)$$

A larger $k$ leads to a lower baud rate but better noise rejection. The switching frequencies in a dc–dc converter are restricted by the hardware parameters, the performance of the controller and the design requirements. Here we chose 100 and 83.3 kHz (with periods of 10 and 12 µs, respectively) as the switching frequencies $f_1$ and $f_0$. Since the power line was multiplexed as a communication channel and thus was noisy and uneven, we conservatively selected a constant of $k = 6$; thus the baud rate was 2.78 kB (with a period of 240 µs), which is fast enough for the transmission of state and control information. These switching frequencies and baud rate are not unique; there are other options. We adopted a TI TMS320F28035 digital signal processor (DSP) as the controller for both the buck/boost converter and the receiver.

The receiver parameters are listed in Table 2. A signal conditioning circuit extracts the switching frequency component, which is superimposed with high-order harmonics of the switching frequency and noise in the power line. This circuit consists of a dc blocking capacitor, a differential amplifier, an active second-order high-pass filter, an active second-order LPF and a dc voltage bias circuit, as shown in Fig. 7d. Only the switching frequency component remains in the output after the DFT demodulation performed in the DSP.

The general DFT algorithm is expressed as follows:

$$X(k) = \sum_{n=0}^{N-1} x(n)e^{-j\frac{2\pi}{N}nk} \quad (k = 0, 1, \ldots \ldots, N-1) \quad (6)$$

Under the assumption that the current DFT value is based on a sequence {$x(0), x(1),\ldots x(N-1)$} and the next is based on a sequence {$x(1), x(2),\ldots x(N)$}, an iterative method can be employed to implement a sliding DFT; then the new DFT value after the next sample is:

$$X(k)_{new} = [X(k) - x(0) + x(N)]e^{jk\Omega_0} \quad (7)$$

Equation (7) decreases the DFT calculation time significantly, making it suitable for implementation on most micro-processors, such as a DSP or a field programmable gate array (FPGA) processor[27].

**Experiment with communication protocol**. In any application, a communication protocol above the physical layer is indispensable; therefore, a simple data link protocol was incorporated into the experiment. The data frame structure was defined as follows: 1 start bit, 13 data bits (from low to high), 1 odd parity bit, and 1 end bit. In the idle state, the switching frequency was 100 kHz. The output current of the buck/boost converter was packed into frames and sent to the receiver. The experimental results shown in Fig. 8 indicate that the output current (2.15 A,

---

**Table 1 Experimental parameters of the conventional buck/boost converter.**

| Parameter | Value/type |
|---|---|
| $V_{in}$ (dc input voltage) | 48 V |
| $V_{out}$ (dc output voltage) | 24 V |
| $C_{in}$ (input capacitance) | 1 mF |
| $L$ (inductance) | 650 µH |
| $C_{out}$ (output capacitance) | 1 mF |
| $Z_{load}$ (load resistance) | 50 Ω |
| Micro-controller | Texas Instruments TMS320F28035 |
| MOSFETs | Infineon IPB320N20N3 G |
| $K_p$ | 2 |
| $K_i$ | $1 \times 10^4$ |
| $T_s$ (sampling period) | 20 µs |
| $f_1$ (switching frequency for normal operation and sending data '1') | 100 kHz |
| $f_0$ (switching frequency when sending data '0') | 83.$\dot{3}$ kHz |
| Baud rate | 2.78 kB |
| Bit rate | 2.78 kbps |

---

**Table 2 Experimental receiver parameters for the conventional buck/boost converter.**

| Parameter(s) | Value/type | Parameter(s) | Value/type |
|---|---|---|---|
| Micro-controller | Texas Instruments TMS320F28035 | $R_2$, $R_3$ | 10 kΩ |
| Operational amplifier | LF353 | $R_4$, $R_5$, $R_{11}$ | 100 kΩ |
| Signal sampling frequency | 1 MHz | $R_6$, $R_7$ | 16 kΩ |
| $C_1$, $C_5$ | 100 nF | $R_8$ | 1 kΩ |
| $C_2$ | 10 nF | $R_9$ | 1.3 kΩ |
| $C_3$ | 2.2 nF | $R_{10}$ | 3.6 kΩ |
| $C_4$ | 470 pF | $R_{12}$ | 27 kΩ |
| $R_1$ | 2.2 kΩ | $R_{13}$ | 56 Ω |

oscilloscope measurement) of the buck/boost converter was correctly sent and received.

**Application in LED lighting system**. In the LED lighting system, a photodetector was adopted as the light receiver to extract the signal sent via light, as shown in Fig. 9a–d, and the signal conditioning circuit in the receiver and light receiver are

shown in Fig. 9e. The signal conditioning circuit extracts the switching frequency component, which is superimposed with high-order harmonics of the switching frequency and noise in the power line/visible light channel. This circuit consists of a dc blocking capacitor, a differential amplifier, an active second-order band-pass filter and a dc voltage bias circuit. Only the fundamental component of the switching ripples remains in the output after the DFT demodulation performed in the FPGA. The experimental parameters are listed in Tables 3 and 4. In this experiment, the switching frequency was 1 MHz in the converter, and the

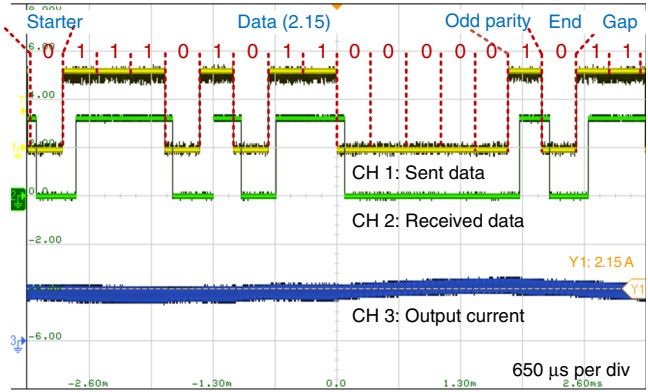

**Fig. 8 Experimental results for a sent and received output current value.** The data frame structure is as follows: 1 start bit, 13 data bits (from low to high), 1 odd parity bit, and 1 end bit. The sent and received data value is 2.15. The bottom blue trace shows the output current of the buck/boost converter, measured as 2.15 A on an oscilloscope.

**Table 3 Experimental parameters of the buck/boost converter in the LED lighting system.**

| Parameter | Value/type |
| --- | --- |
| $V_{in}$ (dc input voltage) | 12 V |
| $V_{out}$ (dc output voltage) | 5 V |
| $C_{in}$ (input capacitance) | 120 µF |
| $L$ (inductance) | 8 µH |
| $C_{out}$ (output capacitance) | 10 µF |
| Load | LED |
| Micro-controller | Cyclone IV EP4CE10F17 |
| MOSFETs | TPH3206 |
| $f_1$ (switching frequency when sending data) | 1 MHz |
| $f_0$ (switching frequency during normal operation) | 0.833 MHz |
| Baud rate | 100 kB |
| Bit rate | 200 kbps |

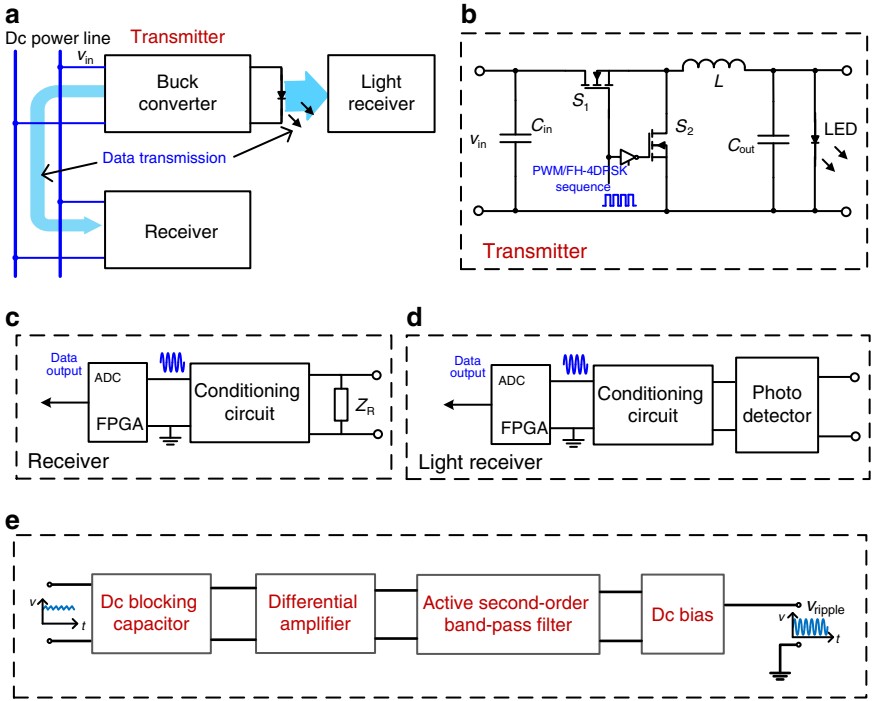

**Fig. 9 Structure of the prototype LED lighting system.** In this experiment, PWM/FH-4DPSK modulation are employed. **a** The general structure of the experimental prototype, which includes a buck/boost converter as a data transmitter, a receiver sharing a common power line with the buck/boost converter and a light receiver. $v_{in}$ denotes the input voltage of the buck converter. Data are transmitted from the buck/boost converter to the receiver via the dc power line and to the light receiver via the visible LED light. **b** Detailed buck/boost converter circuit, in which $v_{in}$ denotes input voltage, $C_{in}$ and $C_{out}$ denote input and output capacitors respectively, and $L$ denotes the inductor. The PWM/FH-4DPSK modulated signal is employed as the gate signal for MOSFET $S_1$, and its complementary signal is the gate signal for MOSFET $S_2$. **c** The structure of the receiver. It consists of an impedance $Z_R$, a signal conditioning circuit and an FPGA with ADC to output digital data. **d** The structure of the light receiver, which consists of a photodetector, a signal conditioning circuit and an FPGA. **e** The signal conditioning circuit used in both the receiver and the light receiver. This circuit is composed of a dc blocking capacitor, a differential amplifier, an active second-order band-pass filter and a dc voltage bias circuit. The input signal is conditioned, and at the output port, the fundamental component of the switching ripple with a proper dc bias is obtained, denoted by $v_{ripple}$.

**Table 4 Experimental parameters of the receiver and light receiver in the LED lighting system.**

| Parameter(s) | Value/type | Parameter(s) | Value/type |
|---|---|---|---|
| Micro-controller | Cyclone IV EP4CE10F17 | $R_3$, $R_4$, $R_5$ | 2 kΩ |
| Operational amplifier | LMH6643 | $R_6$ | 3.9 kΩ |
| Signal sampling frequency | 10 MHz | $R_7$ | 11 kΩ |
| $C_1$, $C_2$, $C_5$ | 100 nF | $R_8$ | 18 kΩ |
| $C_3$, $C_4$ | 80 pF | $R_9$ | 1.8 kΩ |
| $R_1$, $R_2$ | 1 kΩ | $R_{10}$, $R_{11}$ | 10 kΩ |

communication rate of 200 kbps was much higher than that in the first experiment to achieve better talkative power performance.

## Data availability

The source data underlying Fig. 5b, d are available at Figshare: https://figshare.com/articles/Source_Data_rar/11828253. The additional data that support the findings of this study are available from the corresponding author upon reasonable request.

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

## Acknowledgements

This research is supported by National Science Foundation of China (51490682) and Ministry of Science and Technology of China (2017YFE0112400).

## Author contributions

X.H. conceived the idea of integrating power electronics and communication, led the project, participated in paper writing and revision, and provided guidance to all co-authors. R.W. performed the theoretical analysis of the integration method, designed and implemented experiments, and wrote the paper/revision. J.W. provided the ideas and methods on theoretical analysis, experimental validation, and applications and participated in paper writing. W.L. offered idea on the integration method, experimental prototype, and paper writing.

## Competing interests

The authors declare no competing interests.
