## [Peer Review File · Nature Communications]

Reviewers' comments:

Reviewer #1 (Remarks to the Author):

Interesting, creative, novel and informative.

Significant contribution linking power electrical engineering and communications.

Reviewer #2 (Remarks to the Author):

The major claim is the novel frequency-hopping differential phase shift keying (FH DPSK) proposed in this paper to overcome the issue of interference between chosen transmission systems. The abstract must be modified to reflect this proposed novelty.

The claims appear novel and the manuscript reviews what will become an increasingly important area where luminaires are dual-purposed to provide both light and communications.

This is a technology that is going to have general application in the Internet of Things as well as power supply provision in general where clearly innovation is going to be a requirement. The manuscript will certainly be of interest to the readership and may influence the thinking of both communication engineers and power supply engineers. The manuscript is timely given that at the research stage there are likely to be some significant advances in the coming 5-10 year period.

The manuscript will influence thinking by reviewing the current state of the art in the use of LEDs/light to provide bi-directional communications, as well as providing one new advanced communication technique.

The manuscript requires some editing, particularly for language, and this would enhance the manuscript. It would be useful to clearly link the literature review with the innovative FH DPSK keying proposed.

Reviewer #3 (Remarks to the Author):

This paper is dealing with so called power line communication (PLC) technology with DC/DC converter which is one of power electronic circuits. Recently, many distributed generators with power electronic converters are used in power systems, therefore communications between power converters are expected. This paper deals with a timely topic in this trend. The proposed method is new and verified experimentally. Therefore this paper is worth for publishing. Following small corrections will be required.

1) In Fig.4 c and d, V_g will be v_g . Also in the figure caption, "a" in line 211 will be "c" and "b" in line 214 will be "d".

2) In the first line of equation (2), parentheses will be required after the symbol of sigma.
 $\Sigma(e^{-j2\pi f_k T_s} - e^{-j2\pi f_{k+1} T_s})$

Reviewer #1 (Remarks to the Author):

Interesting, creative, novel and informative.

Significant contribution linking power electrical engineering and communications.

Response:

Thanks for your comments. We have revised the manuscript and improved English language according to your suggested version forwarded by the editor.

Reviewer #2 (Remarks to the Author):

The major claim is the novel frequency-hopping differential phase shift keying (FH DPSK) proposed in this paper to overcome the issue of interference between chosen transmission systems. The abstract must be modified to reflect this proposed novelty.

Response:

Thanks for your comments. We have added "A novel modulation strategy called frequency hopping-differential phase shift keying (FH-DPSK) is also presented to overcome the crosstalk between chosen transmission systems." in the revised abstract (highlighted in page 1 of revised manuscript).

The claims appear novel and the manuscript reviews what will become an increasingly important area where luminaires are dual-purposed to provide both light and communications.

This is a technology that is going to have general application in the Internet of Things as well as power supply provision in general where clearly innovation is going to be a requirement. The manuscript will certainly be of interest to the readership and may influence the thinking of both communication engineers and power supply engineers. The manuscript is timely given that at the research stage there are likely to be some significant advances in the coming 5-10 year period.

The manuscript will influence thinking by reviewing the current state of the art in the use of LEDs/light to provide bi-directional communications, as well as providing one new advanced communication technique.

Response:

Thanks for your comments. The distributed power electronic systems, modularized dc devices, and the Internet of Things, which like luminaires, are dual-purposed to provide both energy and communications. The application technology is discussed and novel FH-DPSK modulation proposed with the talkative power converter could be a valuable idea for new advanced techniques in these important areas including conventional LED/lighting technology. Related content is added and highlighted in page 2 of the revised manuscript.

The manuscript requires some editing, particularly for language, and this would enhance the manuscript. It would be useful to clearly link the literature review with the innovative FH DPSK keying proposed.

Response:

Thanks for your comments. We have improved English language in the revised

manuscript with the help from the English native experts and Nature Research Editing Service.

We have considered the visible light communication (VLC) as an example where LED flickering seems one of the challenges according to literature review which are added as references [28]-[30] in the revised manuscript. The novel FH-DPSK modulation with the talkative power strategy is theoretically different from conventional VLC, and there is no LED flickering, as well as no interference between power conversion and data transmission. Related content is added and highlighted in page 10 before the application experimental verification in the revised manuscript.

Reviewer #3 (Remarks to the Author):

This paper is dealing with so called power line communication (PLC) technology with DC/DC converter which is one of power electronic circuits. Recently, many distributed generators with power electronic converters are used in power systems, therefore communications between power converters are expected. This paper deals with a timely topic in this trend. The proposed method is new and verified experimentally. Therefore this paper is worth for publishing. Following small corrections will be required.

1) In Fig.4 c and d, V_g will be v_g . Also in the figure caption, "a" in line 211 will be "c" and "b" in line 214 will be "d".

2) In the first line of equation (2), parentheses will be required after the symbol of sigma.

$$\sum(e^{-j2\pi f_k T_s} - e^{-j2\pi f_{k+1} T_s})$$

Response:

Thanks for your comments. We have modified the symbol and fig. captions mentioned in 1) and 2). The changes are highlighted in Fig. 4 in page 8 and equation (2) in page 3.

REVIEWERS' COMMENTS:

Reviewer #1 (Remarks to the Author):

The language is now, much better.

Reviewer #3 (Remarks to the Author):

This paper is dealing with so called power line communication (PLC) technology with DC/DC converter which is one of power electronic circuits. Recently, many distributed generators with power electronic converters are used in power systems, therefore communications between power converters are expected. This paper deals with a timely topic in this trend.

Toshifumi ISE, Professor Emeritus, Osaka University

Reviewer #1 (Remarks to the Author):

The language is now, much better.

Response:

Thanks for your comments.

Reviewer #3 (Remarks to the Author):

This paper is dealing with so called power line communication (PLC) technology with DC/DC converter which is one of power electronic circuits. Recently, many distributed generators with power electronic converters are used in power systems, therefore communications between power converters are expected. This paper deals with a timely topic in this trend.

Response:

Thanks for your comments.